# Similarities and Differences between Psychosocial Determinants of Bullying and Cyberbullying Perpetration among Polish Adolescents

**DOI:** 10.3390/ijerph20021358

**Published:** 2023-01-11

**Authors:** Marta Malinowska-Cieślik, Dorota Kleszczewska, Anna Dzielska, Monika Ścibor, Joanna Mazur

**Affiliations:** 1Department of Environmental Health, Faculty of Health Sciences, Jagiellonian University Medical College; 31-066 Krakow, Poland; 2Institute of Mother and Child Foundation, 01-211 Warsaw, Poland; 3Department of Child and Adolescent Health, Institute of Mother and Child, 01-211 Warsaw, Poland; 4Department of Humanization in Medicine and Sexology, Collegium Medicum, University of Zielona Gora, 65-729 Zielona Gora, Poland

**Keywords:** adolescent, bullying, cyberbullying, perpetration, psychosocial factors

## Abstract

Despite the extensive literature on the psychosocial determinants of bullying and cyberbullying among adolescents, there is not sufficient comprehensive analysis of the differences between perpetrators. This study aims to assess the psychosocial similarities and differences between bullies and cyberbullies. Data of 3650 students from two grades, K9 and K11 (47% females, mean age = 16.53), were used from a survey conducted in Poland in 2018 as part of the Health Behaviour in School-aged Children study. Perpetration was measured by questions adapted from the Olweus Bully/Victim Questionnaire. The following predictors were analyzed: demographic, socioeconomic status measured with the Family Affluence Scale-III (FAS), and individual and social factors. Multiple binary logistic regression was applied. The random sampling design was based on stratification according to the local deprivation index of the region where the school was located. Male gender, younger age, and non-intact family were associated with a higher risk of bullying and cyberbullying. Family support, empathy, school performance, and school attachment had a protective effect against both aggressive behaviors. Life dissatisfaction and high FAS were revealed as determinants of cyberbullying and local deprivation of bullying only. Bullying and cyberbullying school prevention programs should take into account these psychosocial differences and consider the economic deprivation of the region.

## 1. Introduction

Peer interpersonal violence among teenagers poses a significant public health issue due to health-related and socioeconomic consequences. In-person bullying and cyberbullying represent the most common forms of peer aggression in adolescence. 

Bullying is defined as a series of repeated, unwanted, aggressive acts intended to harm or intimidate someone less powerful. It is to be understood as physical, verbal, and relational behavior that involves one party having the intention to hurt or harm another within an uneven relationship of power where the victim is unable to defend him- or herself. It is to be understood as a specific type of aggression in which the behavior is intended to harm or disturb, the behavior often occurs repeatedly over time, and there is an imbalance of power, with a more powerful person or group attacking a less powerful one [1]. Cyberbullying can be defined as an intentional behavior aimed at harming another person or persons by means of electronic devices, such as computers, mobile phones, and others. It is typically described as aggression that is intentionally and repeatedly carried out in an electronic context (e.g., e-mails, blogs, pictures, text messages) against a person who cannot easily defend him- or herself and is perceived by him or her as an aversion [2]. Peer violence in adolescents may carry severe mental and physical health consequences not only for the victims but also for the perpetrators who often exhibit other antisocial and health-risk behaviors. Evidence shows that youth who bully others are also at risk of short- and long-term emotional and behavioral problems that continue into adulthood [3]. Compared with those not involved, perpetrators experience more emotional and psychosomatic problems and do not feel safe and cared for at home and in the school environment [4,5]. However, some studies suggest both negative and ‘positive’ outcomes of perpetration, depending on the intentionality of violence and congruency with group norms [6]. 

Even though an overlap between bullying and cyberbullying among adolescents has been proven, there is an ongoing discussion concerning the extent of the impacts and effects of psychosocial determinants. One theory postulates that these are different phenomena, whereas others define cyberbullying as bullying in cyberspace. Cyberbullying is more likely to occur among in-person bullies, and bullies tend to engage in cyberbullying more often than non-bullies [7,8]. Although some researchers say that cyberbullying is a new form of peer aggression, others emphasize that the co-occurrence of these phenomena does not necessarily indicate conceptual convergence. This may overlap appearing in different contexts, both in-person and in cyberspace. 

There are two distinct perspectives that examine the relationship between face-to-face bullying and cyberbullying among adolescents. The first perspective focuses on the differences and suggests that the two types of bullying are qualitatively distinct behaviors. The second approach posits that cyberbullying is an extension or a new form of ‘traditional’ bullying and focuses on the similarities. These perspectives have implications for bullying and cyberbullying research and prevention [9]. A relatively high level of overlap has been demonstrated between face-to-face victimization and cyber victimization [10], but wide country context variations are observed [11]. At the same time, studies have found relatively little overlap between in-person and cyber perpetration. This may indicate that the two may be separate phenomena stemming from different mechanisms, as has been shown in a study in the context of Nordic countries [12].

Demographic factors such as gender and age have received special attention in the literature devoted to the prevalence of bullying and cyberbullying [13,14]. The findings of a systematic review by Zych et al. [5] showed that one out of every three adolescents is involved in bullying in some way, and one out of every five is involved in cyberbullying. There is a clear gender difference in the trends of bullying perpetration, with the rates for boys being higher than those for girls in most European countries, including Poland [15]. Boys are usually more involved than girls but with small or trivial effect sizes. However, the prevalence of bullying others varies considerably across Europe. A study of Swedish adolescents showed discrepant gender patterns of involvement in face-to-face bullying and cyberbullying. Boys were more likely to be face-to-face bullies, and girls were as likely as boys to be cyberbullies. Compared with in-person bullying, girls were generally more involved in cyberbullying relative to boys [16]. As far as age is concerned, school bullying perpetration decreases in late adolescence, whereas cyberbullying rates appear to increase [17]. According to the report of the Health Behaviour in School-Aged Children (HBSC) 2017/18 survey conducted in 45 European countries and regions as well as Canada among adolescents between 11 and 15 years old, in most countries, bullying perpetrator rates increased to a relatively small degree with age, and the relationship with age was weak [18]. In a combined international sample of the HBSC, the prevalence of bullying and cyberbullying was differentiated according to gender. 

Regarding international rankings and comparisons of bullying perpetration, Polish teenagers occupied a middle position. Among 15-year-old boys the percentage of bullies was similar to the HBSC average. Girls were less likely to be perpetrators both in Poland and across the whole combined international HBSC sample. Among boys, 10% were perpetrators compared with the 9% HBSC average, and Polish girls bullied others less often (5%) compared with 9% the HBSC average. Unfortunately, regarding cyberbullying, Polish teenagers occupied the very unfavorable fourth position in the international HBSC ranking, making Poland one of the countries with the highest percentage of young people who commit online aggression. Among 15-year-olds, as many as 24% of Polish boys and 16% of girls cyberbullied others compared with 14% of boys and 8% of girls as the HBSC average. 

Family structure is another important determinant of adolescent involvement in aggressive behaviors. Studies have documented that stressful changes in family life, such as divorce or separation, can result in negative antisocial behaviors and peer aggression. One study showed that adolescents from divorced or remarried families were at risk of exhibiting aggression, at a moderately increased risk of displaying antisocial behaviors, and were more likely to be involved in bullying as compared with adolescents from intact families [19].

Considering social structural factors, the socioeconomic deprivation of the region and inequalities between groups in the community impacts interpersonal violence among adolescents. Epidemiological studies have identified links between bullying and area-level income and showed that income inequality was associated with higher rates of bullying [20]. If we consider the microsystem in which the young person lives, impoverished communities are predictive of antisocial behavior. Studies indicate that the differential effect of impulsivity on violence can be attributed to both developmental processes that lead to a greater concentration of violent and impulsive adolescents in economically deprived neighborhoods as well as a greater likelihood of impulsive adolescents engaging in violence when they reside in economically disadvantaged communities. Young people from less privileged places of residence tend to become involved in violent and aggressive behavior and tend to become perpetrators of bullying more often than their peers living in wealthier regions [21,22]. There is evidence that school-level deprivation influences school bullying. School-level deprivation and low socio-economic status (SES) of the school’s neighborhood are associated with higher rates of bullying among students. A UK study showed that pupils from more deprived areas and schools with a lower quality rating were more likely to report being perpetrators of cyberbullying than students from mainstream academies [23]. 

Considering the relationship between SES of the family and bullying perpetration, studies show varying results. A Finnish study showed that bullying others was associated with low SES, as measured by economic disadvantage, poverty, and low parental education. Students from low socioeconomic backgrounds have been found to bully others more often, and the likelihood of bullying increased markedly among adolescents with the most socioeconomic adversities [24]. However, another study conducted among adolescents from Nordic countries showed that higher family affluence was related to in-person bullying, while it was unrelated to the risk of cyberbullying [12].

To understand the psychosocial determinants of adolescent violence, the social ecological model (SEM) is widely applied [25]. The SEM application is helpful for understanding the possible predictors of bullying and cyberbullying related to society, community, school, family, peers, and individuals [26]. It allows researchers to address the factors that protect young people from perpetrating violence as well as prevention strategies that can be used. Determining the relative strength of individual and contextual predictors is useful for identifying targets for prevention and interventions in public health policies. 

Many studies focused on identifying individual and social risk factors, but potential protective factors of youth violence have been identified as well [21,27,28,29]. Studies pay attention to the protective role of individual social and emotional competencies, such as empathy, self-efficacy, life satisfaction, and problem solving, and good academic performance and strong bonding to school were related to low bullying and cyberbullying perpetration [4,30,31,32,33]. Studies that review the psychosocial correlates between bullying and cyberbullying have highlighted the importance of empathy. Empathy has consistently been found to have a strong negative relationship with antisocial behaviors, and it was established to be related to low rates of in-person and cyber-perpetration in many studies [34,35,36]. A critical review and meta-analysis of cyberbullying research conducted by Kowalski [4] showed that high life satisfaction was associated with low cyber-perpetration, while these relationships are unclear in face-to-face bullying [37]. Besides the number of individual determinants of violence, the SEM draws attention to the social microsystem, which includes the family, peers, and school [38,39]. A high quality of relationships and communication in the family has been indicated as a strong protector. Overall, a positive home environment is related to low bullying and cyberbullying, and parental support and attachment have been revealed as the strongest preventive factors [36,40,41,42]. As far as peer variables are concerned, studies identified peer support, positive peer relationships, and friendships as protective factors against violence among adolescents [4,29]. Regarding school, bonding to school and alleviating difficulties in school stress management prevent involvement in peer violence. It has been proven that a positive social school climate and satisfaction with interpersonal relationships at school were related to lower rates of bullying and cyberbullying perpetration [31,36,42]. In addition to the interaction of the individual and social factors discussed above, the SEM includes the wider structural determinants of violence among adolescents, which are related to socioeconomic and political contexts, such as local population income, labor, living conditions, access to health and social care, and access to education in the region [38,39]. 

Given that bullying and cyberbullying are both very complex phenomena, this study analyzes protective factors taking into account a dynamic interplay of different variables and levels of analysis according to the SEM perspective. It highlights the important and interrelated protectors at the individual level (e.g., academic performance, social self-efficacy, life satisfaction, and empathy), the close relationships and social circles such as peers and family (e.g., family and peer support), and the school setting where social relationships occur (e.g., school attachment). According to our knowledge, these psychosocial factors have not been taken into account when comparing the perpetrators of these two forms of peer violence in later adolescence. Our research, including a group of 17-year-olds, supplements the state of knowledge related to the HBSC studies, which show the trends concerning adolescent violence up to the age of 15.

There is a controversy pertaining to whether cyberbullying is simply a form of bullying that occurs in cyberspace. Even though they overlap, we try to answer whether these two peer-violence-related behaviors should be analyzed as distinct phenomena, taking into account differences in the strength of psychosocial protective factors. Despite the extensive literature regarding psychosocial determinants of bullying and cyberbullying, we identified a shortage of more comprehensive comparisons and analyses of the differences between these two groups of perpetrators. Therefore, we aimed to assess the extent to which the psychosocial variables that have been found as potential protective factors predict the perpetration of in-person bullying and cyberbullying among adolescents. The objective of this research was to assess whether bullies and cyberbullies share common predictors or whether they can be differentiated by unique patterns. We wanted to assess the extent to which bullies and cyberbullies differ by demographic, socioeconomic, and psychosocial factors. Accordingly, we posed three research questions.

*RQ1:* What is the prevalence of bullying and cyberbullying among Polish adolescents? 

*RQ2:* To what extent does the perpetration of bullying and cyberbullying overlap? 

*RQ3:* What are the similarities and differences between bullies and cyberbullies?

## 2. Materials and Methods

### 2.1. Sample and Procedure

The study used data from a cross-sectional survey implemented in Poland as part of the HBSC research project conducted in 2017/2018. A standardized and validated questionnaire was applied [15]. The translation procedure for the questionnaire followed an international HBSC survey protocol. Depending on the reviewer’s comments, the translations were reviewed, modified, and finally accepted [43]. 

The study scheme applied stratified cluster sampling with classes within schools as the primary sampling unit. The schools were randomly selected, and individual classes in these schools were subsequently randomly admitted. The design considered stratification according to the local deprivation (LD) index, which means that schools from regions with different levels of economic development were reached. The LD index is a composite indicator that measures a region’s level of development based on external socioeconomic data. It has been defined as a combination of five sub-indices for population income, employment, living conditions, education, and access to goods and services [44]. 

Self-completion, paper-printed questionnaires were administered in the classroom in the presence of trained interviewers that were responsible for data collection. Consent was obtained from school directors, parents or caregivers, and pupils. The principals of the schools and local education authorities were informed about the survey. The adolescents’ participation was anonymous and voluntary, and no incentives were offered. The research received approval from the Bioethical Commission operating at the Institute of Mother and Child in Warsaw (No. 17/2017 with Annex 1, dated 30.03.2017). 

The study sample consisted of 3650 adolescents (47% females). Bullying perpetration data analysis was based on answers obtained from 3650 students who responded to the bullying questions, while 13 (0.4%) were missing data. Cyberbullying perpetration data analysis was based on answers from 3643 pupils who responded to the cyberbullying questions, while 7 (0.2%) were missing data. The data were collected from students of two grades: K9 and K11 (53.6% of students of K9, and 46.4% of K11 grade). These two age groups among four collected in the Polish HBSC survey, and students of K11 grade were studied outside the international HBSC protocol. The mean age was 16.53 years (SD = 1.09). For grade K9, the mean age was 15.57 years (SD = 0.39), and for grade K11, the mean age was 17.63 years (SD = 0.36).

Among the respondents, 75.3% lived in an intact family, while 24.7% lived in a non-intact family. The socioeconomic status of the participant’s family was measured by the HBSC Family Affluence Scale-III (FAS-III) with value ranging from 0 to 13. The mean family affluence was 7.83 (SD = 2.26). The proportions of study sample students from regions classified into the successive quintiles (Q) of local deprivation, where Q1 denotes the poorest and Q5 denotes the richest regions, were: Q1—13.3%; Q2—13.4%; Q3—16.5%; Q4—18.4%, and Q5—38.4%. Regarding place of residence, 40.8% of respondents lived in rural areas, whereas 59.2% lived in urban areas. Of those in urban areas, 25.4% lived in cities with a population of more than 100,000 inhabitants.

### 2.2. Measures

#### 2.2.1. Bullying and Cyberbullying Perpetration

Outcome variables were measured using a single question for in-person bullying and a single question for cyberbullying perpetration using a modified and adapted version of the Olweus Bullying Questionnaire [45,46]. The students were asked ‘How often have you taken part in bullying another student(s) at school in the last two months?’ The question was preceded by a short explanation of what was regarded as bullying. The participants answered to indicate whether they had perpetrated bullying in the last two months prior to the administration of the survey, with five response categories ranging from ‘I have not bullied in this way in the past two months’ to ‘several times a week’. Five ordinal response categories were dichotomized into a binary outcome (‘never’ vs. ‘at least once in the last two months’). Responses were recoded as 0 (never) and 1 (once or more). Regarding cyberbullying, the participants indicated how often they had taken part in cyberbullying in the last two months prior to the administration of the survey, i.e., how often they had: sent mean instant messages, emails, or text messages; made wall postings; created a website making fun of someone; posted unflattering or inappropriate pictures online without permission or shared them with others. Its five ordinal response categories ranging from ‘I have not cyberbullied in this way in the past two months’ to ‘several times a week’ were dichotomized into a binary outcome and recoded as 0 (never) and 1 (at least once or more).

#### 2.2.2. Socioeconomic Factors

Socioeconomic status was assessed using the HBSC Family Affluence Scale-III (FAS III), which is a six-item measure of material assets in the household, including number of vehicles, bedroom sharing, computer ownership, bathrooms at home, dishwashers at home, and family vacations [47,48]. The item scores were summed to a value ranging from 0 to 13 and categorized into three groups: low/least affluent (0–5), average (6–9), and high/most affluent (10–13). In correlation analyses, the FAS-III variable was used as a continuous variable. Even if Cronbach’s alpha for FAS-III is less than 0.7 (reliability index Cronbach’s alpha = 0.544), this tool is recommended for use in HBSC studies. The scale was estimated with Samejima’s graded response model and tested for differential item functioning according to country, including Poland, under the last FAS-III scale validation study. In this study, the test–retest reliability for Poland was r = 0.91, and the FAS correlated with the family income reported by parents with an Eta2 close to 0.30.

The LD index is a composite indicator that measures a region’s level of development based on external socioeconomic data. It has been defined as a combination of sub-indices for: population income, labor, living conditions, education, and access to goods and services [44]. In the case of the local deprivation index, division into five classes has been used, where Q1 denotes the poorest and Q5 denotes the richest school location region.

#### 2.2.3. Individual Factors

School performance (SP) was assessed by self-rating one’s own academic status compared to classmates via an 11-rung ladder ranging from 0 points (the worst) to 10 points (the best academic achievements) [49]. The item scores were categorized into three groups: low (0–4), average (5–7), and high (8–10).

Social self-efficacy (SSE) was measured using subscales of the Self-Efficacy Questionnaire for Children (SEQ-C) by Muris [50]. This scale consists of eight items relating to how well adolescents perceived their own interpersonal communication skills, with responses rated on a five-point scale from 0 (not at all) to 4 (very well). The item scores were summed to a value ranging from 0 to 32 and were categorized according to three levels: low (0–16), average (17–25), and high (26–32). The internal consistency of the scale was very good (Cronbach’s alpha = 0.841), and this confirmed the reliability of this approach.

Empathy was measured using subscales of the Social and Emotional Health Survey by Furlong [51]. This scale includes the following statements: ‘I feel bad when someone gets their feelings hurt’, ‘I try to understand what other people go through’, and ‘I try to understand how other people feel and think’. Responses were rated on a four-point scale from 0 (‘not like me’) to 3 (‘very much like me’). The item scores were summed to a value ranging from 0 to 9 and categorized according to three levels: low (0–4), average (5–7), and high (8–9). The internal consistency of the scale was very good (Cronbach’s alpha = 0.825).

Life satisfaction was measured using the adapted self-rating Cantril’s Ladder of Life Satisfaction [52]. Respondents were asked to indicate where they felt they currently stood on an 11-rung ladder representing a scale from 0 points (‘worst possible life’) to 10 points (‘best possible life”). The item scores were categorized into three groups: low (0–5), average (6–8), and high (9–10).

#### 2.2.4. Social Factors

Family and peer support was assessed using the subscales of the Multidimensional Scale of Perceived Social Support (MSPSS) [53]. Parental support was assessed using an item that asked about relationships and communication with parents that included statements describing help and emotional support, communication of problems, and assistance in decision making. Participants were asked to rate four statements on a five-point Likert scale. The total score ranged from 0 to 24, and an outcome of 0 to12 was categorized as weak; a score of 13 to 22 was considered average; and a total score of 23 to 24 was recorded as strong support. The parent support scales showed very good internal consistency (Cronbach’s alpha = 0.941). The presence of supportive peers was assessed using a four-item scale describing the degree of help from friends, ability to count on them, and communication of feelings and problems. The respondents were asked to answer on a five-point Likert scale. The total score ranged from 0 to 24 and was categorized as weak (0–9), average (10–19), or strong (20–24). The scale showed very good internal consistency (Cronbach’s alpha = 0.895). School attachment was assessed by a single item gauging the students’ emotional and psychological connectedness to school in terms of how much they liked school. The school satisfaction scale was validated and included in the HBSC study protocol for the 2017/2018 survey [43]. The participants answered the question to indicate how they felt about school at the moment, with four possible responses ranging from 1 (‘I don’t like it at all’) to 4 (‘I like it a lot’). The responses were categorized into three levels of school satisfaction: very low; rather low; average or high. 

### 2.3. Statistical Analysis

At the initial stage, summary indexes were calculated for scales relating to family affluence, social self-efficacy, empathy, and parent and peer support. The reliability of these scales was checked using Cronbach’s coefficients. The summary indexes and single items were categorized into three ranges according to the principles described earlier. Response categories for all variables are presented as the number of respondents along with the corresponding percentage. The prevalence of the perpetration of bullying and cyberbullying in groups distinguished according to demographic, socioeconomic, individual, and social factors was compared using a chi-square test. A bivariate Spearman (rho) correlations matrix was created to describe the relationship between the nine potential determinants of bullying and cyberbullying (five summary indices and four original single items). Multiple binary logistic regression analysis was applied to estimate the odds ratio (OR) for becoming a perpetrator of bullying and cyberbullying on predictor variables. The ORs were presented together with the 95% confidence intervals (CI). To assess the goodness-of-fit of the presented models, the pseudo-R-sq Nagelkerke coefficient was applied. Additional models estimated separately for boys and girls are attached as Appendix A. All statistical analyses were performed using the SPSS software, version 27.0 (Statistical Package for Social Sciences, SPSS Inc., Chicago, IL, USA). The level of significance was set at *p* < 0.05.

## 3. Results

### 3.1. Bullying and Cyberbullying Perpetration According to Demographic and Socioeconomic Factors

The study sample included 3650 adolescents who answered bullying questions and 3643 who answered cyberbullying questions. There are different numbers in these independent variables due to missing data. In the total study sample, 22% bullied and 17.2% cyberbullied others at least once in the last two months prior to the administration of the survey. In study sample, cyber and school bullying perpetration overlapped, albeit marginally, more so in boys, and in the younger group. We found that 291 teenagers, that is 8% of the study sample, were perpetrators of both forms of aggression. The rate of face-to-face and cyber-perpetrators was 10.9% in boys and 5.4% in girls; as well as 9.9% in younger (K9) and 5.9% in older pupils (K11).

The percentages of students who bullied or cyberbullied others were analyzed in the context of nine variables revealed as psychosocial predictors of these peer-violence-related behaviors. The variables were classified into three sets: demographic and socioeconomic, individual, and social factors. Table 1, Table 2, Table 3 summarize the descriptive statistics. 

Differences in bullying and cyberbullying according to gender, grade, and family structure were evident, and the rates varied substantially in these groups. FAS did not affect the prevalence of either of these aggressive behaviors in the study sample. LD impacted the prevalence of bullying perpetration, while it had no such effect on cyberbullying (Table 1). 

The results outline variations in perpetration of face-to-face bullying and cyberbullying according to gender and age. Boys perpetrated bullying and cyberbullying more often than girls. In the total sample, the percentage of boys who bullied was 10.8% higher than that of girls and, similarly, was 10.6% higher regarding perpetration of cyberbullying. With respect to age, younger adolescents bullied and cyberbullied more often than older ones, and the difference was larger for face-to-face bullying. Younger pupils (K9) bullied 9.1% more often than older ones (K11), and they cyberbullied 4.5% more often than older pupils (K11). Considering family structure, a similar difference in bullying (4.6%) and cyberbullying (4.2%) was observed, with lower rates revealed among students growing up with both biological parents.

### 3.2. Prevalence of Bullying and Cyberbullying Perpetration According to Psychosocial Factors

The analysis of the relationship between bullying and between cyberbullying and individual factors (Table 2) showed significant differences for both types of behavior in the percentages of students who had a low, average, or high level of empathy and school achievement. Social self-efficacy (SSE) and life satisfaction (LS) were significant in students who perpetrated cyberbullying only.

The percentage of face-to-face and cyber-perpetrators was higher among students with low empathy compared with those with a higher level. The percentage of teenagers who perpetrated both bullying and cyberbullying less often increased in line with improvement in school achievement and empathy level. Among individual factors related to the perpetration of both behaviors, empathy had the most extensive differentiating effect. Pupils with a high level of empathy perpetrated face-to-face bullying 17.6% and cyber-perpetrated 14.4% less often than those with a low level. A similar pattern to that applicable to empathy was observed regarding school performance. Students with high school performance perpetrated bullying 7.5% and, similarly, cyber-perpetrated 7.4% less often than those with low school performance. As school achievement improved, the percentage of students who perpetrated less increased. Stronger SSE and higher LS resulted in a decrease in cyberbullying only. Students with high LS cyber-perpetrated 7.1% less often than those with low LS. A similar pattern was noticed regarding SSE. Pupils with high SSE cyber-perpetrated 4.9% less often than those with low SSE. 

Table 3 shows the percentages of adolescents who bullied and cyberbullied taking social factors into account. A relationship was confirmed for all three variables: family support, peer support, and school attachment. Among students with strong family support, the proportion of those who perpetrated face-to-face bullying was 9.4% lower than among those with low parental support. Similarly, for students who cyberbullied, the proportion was 9.7% lower as compared with those with low family support. As far as peer support is concerned, a more pronounced difference was observed between bullies and cyberbullies. In students with strong peer support, the proportion of those who were involved in bullying was 10.3% lower than among those with weak peer support. Concerning students who cyberbullied, those with high peer support cyber-perpetrated 6.4% less often as compared with those with low peer support. 

The proportion of bullies and cyberbullies was higher among students with very low school attachment compared with those with average or high scores of bonding with school (10.9% and 13.1%, respectively). 

### 3.3. Correlations between the Psychosocial Determinants of Bullying and Cyberbullying Perpetration

Table 4 shows the correlation of nine factors that may potentially impact perpetration of bullying and cyberbullying among teenagers. No correlation was detected in five cases, including four relationships with LD. Furthermore, no correlation was observed between LD and school performance (rho = 0.023; *p* = 0.166), LD and life satisfaction (rho = −0.026; *p* = 0.111), LD and family support (rho = −0.028; *p* = 0.096), or LD and school attachment (rho = 0.028; *p* = 0.092). There was also no correlation between empathy and life satisfaction (rho = −0.032; *p* = 0.056). The strongest correlation was recorded for peer support with social self-efficacy (rho = 0.442; *p* < 0.001), and family support with life satisfaction (rho = 0.415; *p* < 0.001). Statistically significant correlations were recorded for these relationships. Given the values of these coefficients, we checked the assumptions regarding multicollinearity to provide for regression type analysis that the models were valid [54,55]. Because the values of the correlation coefficients between the independent variables used were relatively low, according to general assumptions, this allowed regression analyses to be performed and to estimate the models.

### 3.4. Predictors of Bullying and Cyberbullying Perpetration: Logistic Regression

To compare the psychosocial predictors of bullying and cyberbullying perpetration, multiple binary logistic regression models were estimated on the basis of the Nagelkerke’s R-sq goodness-of-fit statistics. The results for models with bullying and cyberbullying are presented in Table 5 and Table 6. Table 5 shows odds ratios (ORs) for the model with bullying, and Table 6 shows ORs for the model with cyberbullying perpetration. They include nine variables grouped according to demographic (gender, grade, and family structure), socioeconomic (family affluence and local deprivation), and psychosocial factors (school performance, social self-efficacy, empathy, life satisfaction, family support, peer support, and school attachment). The conducted analyses allowed a comparison of models for bullying and cyberbullying perpetration.

The strongest predictor of bullying perpetration was male gender, with the risk of bullying for males amounting to twice the value of the female gender (OR = 1.937). The age of the participants (the grade they attended) also affected bullying to a significant extent, with the risk increasing 1.769 times in younger adolescents in comparison with older ones. Regarding family structure, a 1.230 higher risk of bullying was seen among adolescents from non-intact families compared with those from intact families.

Family support was found to be a very strong predictor of bullying. Compared with adolescents with strong support, the risk of bullying for those with weak support increased 1.873 times, and it increased 1.516 times for those with an average level of support.

Empathy was determined to be almost as strong as family support as a predictor of bullying. In adolescents with low empathy, the risk increased 1.873 times, and it increased 1.352 times for those with an average level of empathy as compared with teenagers with high empathy.

School attachment and school performance were also identified as important predictors of bullying. Pupils who disliked school presented a 1.474 times greater risk of bullying in comparison with pupils who liked school. Similarly, those with low academic achievement presented a 1.321 times greater risk as compared with those with high school performance.

Among socioeconomic factors, local deprivation was established as a predictor in the model for bullying. The risk of bullying among students from schools located in poorer areas (Q1, Q2, and Q4) was higher than among those in rich regions with the highest index (Q5); the risk was 1.370, 1.515, and 1.346 times higher, respectively. 

Regarding cyberbullying, the effect of gender was similar to the bullying model. The male gender was associated with a risk more than twice as high as the female gender (OR = 2.180). Compared with bullying, a lower impact of the age of the participants (the grade attended) was observed, with the risk increasing 1.413 times in younger adolescents in comparison with older ones. Family structure was determined to be a significant factor as well, and there was a higher risk of cyberbullying in students from non-intact families relative to those from families with two biological parents (OR = 1.309). Similar to the model for bullying, results of the model for cyberbullying showed a very strong effect of family support. The risk of cyberbullying increased 1.909 times in adolescents with a weak level of support, and it increased 1.464 times with an average level of support as compared with those benefitting from strong family support. 

The same applies to empathy. In adolescents with a low level of empathy, as compared with those with a high level, the risk of cyberbullying perpetration increased 1.863 times, and the influence of the average level persisted (OR = 1.437). Likewise, in the model for bullying, a significant impact of school attachment was visible (OR = 1.888), which was higher than in the case of the bullying model. A similar impact of school performance was noticed (OR = 1.349) as well. However, unlike the bullying model, a significant effect of dissatisfaction with life was seen in the cyberbullying model, with the risk OR = 1.422 times higher than in adolescents with high life satisfaction. 

Changes in the impacts of socioeconomic factors such as local deprivation and family affluence were identified. In the model for cyberbullying, local deprivation in the place of residence had no effect, whereas an effect of family affluence was revealed. Pupils from highly affluent families were at 1.377 times higher risk of cyber-perpetration as compared with those from families with a low status. 

### 3.5. Gender-Specific Models

In Appendix A, the results of the logistic regression of the gender-specific extended models for bullying and cyberbullying are presented. They allow comparison of the psychosocial factors that determine becoming a perpetrator of these two forms of peer aggression in both genders separately.

In girls (Appendix A), lower levels of empathy and poorer academic performance were predictors of risk of both forms of aggression. An already low or average level of empathy and poor or average academic performance increased the risk of cyberbullying, whereas in the case of face-to-face bullying, only a low level of empathy and poor academic performance were risk factors. Life dissatisfaction and poor attachment to school turned out to be predictors of cyberbullying only. Another factor that increases the risk of in-person bullying and that was irrelevant in relation to cyberbullying was low support from the family.

In boys (Appendix A), low level of empathy and poor school attachment as well as average or poor family support were predictors of both forms of peer aggression. Younger age and lower index of local deprivation increased the risk of in-person bullying. At the same time, these factors were not relevant to cyberbullying.

## 4. Discussion

Both in-person and cyberbullying remain prevalent problems that negatively affect all youth involved, including the perpetrators. This study draws attention to the differences and similarities between bullies and cyberbullies in relation to the demographic, socioeconomic, and psychosocial determinants of peer aggression among adolescents. Whereas risk and protective factors have been studied extensively [4,29], we found that the impact of these factors on students perpetrating in-person bullying and cyberbullying still needs to be assessed and compared, and the differences and similarities in predictors for bullying and cyberbullying need to be identified. Based on a literature review, gender, age, family structure, and nine other psychosocial predictors were selected and grouped into three substantive sets of socioeconomic, individual, and social factors taking into account the Social-Ecological Model (SEM) [39,42].

The objective of this research was to assess whether bullies and cyberbullies share common predictors or whether they can be differentiated by unique patterns of these determinants. The analysis was guided by three research questions formulated in the introduction. Our findings highlight shared and unique psychosocial predictors that protect adolescents from both forms of peer aggression. The application of multiple binary logistic regression enabled us to assess the impact of these predictors and estimate the risk of bullying and cyberbullying perpetration. 

The overlap between bullying and cyberbullying among adolescents has been proven; however, the extent of the impacts of different psychosocial determinants are still subject to discussion. Some theories postulate that these are different phenomena, while others define cyberbullying as bullying in cyberspace. Although some researchers say that cyberbullying is a new form of peer aggression, others emphasize that the co-occurrence of these phenomena does not necessarily indicate conceptual convergence. This may overlap appearing in different contexts, including both in-person and in cyberspace [9]. Studies have documented the overlap between bullying victimization and cyber-victimization, but wide country context variations are observed [11]. As far as perpetration is concerned, cyberbullying is more likely to occur among in-person bullies, and bullies tend to engage in cyberbullying more often than non-bullies [5,7,8]. Our study found relatively little overlap between these two behaviors. This may indicate that the two are separate phenomena stemming from different mechanisms, as has been shown in the Nordic context [12]. In our study, cyber and school bullying perpetration co-occurred to a relatively low extent. Perpetrators of both forms of aggression accounted for 8.0% of the study sample; prevalence was higher in boys (10.9%) than girls (5.4%) and in younger (9.9%) more than older students (5.9%).

In the literature devoted to the perpetration of bullying and cyberbullying, demographic factors have received special attention. Our results outlined variations in the perpetration of bullying and cyberbullying and differ according to gender and age. Boys perpetrated bullying and cyberbullying more often than girls, and younger adolescents were perpetrators more often than older ones. Studies have shown discrepant gender patterns of involvement in bullying and cyberbullying. In a study of Swedish adolescents, boys were more likely to be in-person bullies, and girls were equally likely as boys to be cyberbullies [16]. Other studies have shown that bullying and cyberbullying perpetration is more common in boys than in girls [56]. Regarding age, studies show that school bullying perpetration decreases in late adolescence, whereas cyberbullying rates appear to increase [17]. As far as cyberbullying is concerned, the results of a meta-analysis conducted by Kowalski et al. [4] showed a weak observed correlation for cyberbullying perpetration and age. In the HBSC studies among 11-, 13-, and 15-year-olds, in-person bullying rates decreased with age in about two thirds of the countries [57]. The report from Polish research connected with the HBSC survey conducted on an extended age group of students from 11 to 17 years old tracked age-related changes an found an increase in bullying rates in the group of 13-year-olds followed by a decrease in the older age groups (15- and 17-year-olds). This was observed both in boys and in girls. Regarding cyberbullies, increasing rates were seen in the group of 15-year-olds, which was followed by a decrease in rates in 17-year-olds. In that case, the downward trend was more pronounced among 17-years-old girls [58]. Our study showed a higher prevalence of bullying in the younger group. The findings confirm that younger students from K9 grade were found to be more at risk than those from K11 both in bullying and cyberbullying. A decrease in both types of aggressive behaviors was observed. We established that adolescents living in non-intact families were more at risk of becoming a bully or a cyberbully than those living with both biological parents. This is in line with the results of other studies, which showed that adolescents from non-intact families exhibited higher levels of aggression, antisocial classroom conduct, and decreased level of emotional self-regulation [19].

Findings regarding the impact of socio-economic factors on the perpetration of bullying and cyberbullying deserve special attention as well. Our study supported the association between youth bullying and the socioeconomic status of the community as measured using the local deprivation index. Although a protective effect lower LD index (higher wealth of the region) was revealed in bullying only and was not significant in cyberbullying, it is worth considering LD in prevention. Living in a richer and wealthier region protected students from becoming an in-person bully. This could be explained by the results of another study, which showed that socioeconomic disadvantage in the community shapes the development of impulsivity and provides a situational context for tendencies to manifest in violent and aggressive behaviors among adolescents [22,59]. Knaappila et al., confirmed that the overall likelihood of bullying increased significantly among adolescents facing numerous socioeconomic adversities, low family SES, and inequalities [24]. A study of adolescents from Nordic countries showed that higher family affluence was unrelated to the risk of cyberbullying; however, it was related to in-person bullying and combined forms of bullying [12]. Our findings have not confirmed the association between family FAS and bullying. It could be explained by the findings of the meta-analysis study by Tippett et al. [60] indicating the problem of standard measurements of SES of the family and the use of different indicators, which show varying effects.

However, higher FAS appeared to be a risk factor for cyber-perpetration. Living in a wealthier family increased the risk of becoming a cyberbully, and adolescents from wealthier families cyberbullied others more often than those from less affluent families. Studies have shown that accessibility of social media leads to new opportunities for online aggression, and low frequency of ITC technology use was found to be related to lower cyber perpetration rates [2,29,42,61]. Our results may suggest that cyberbullying is associated with greater access and exposure to computers, mobiles, and the Internet among adolescents from highly affluent families. However, more research on this subject is required.

Regarding psychosocial factors, a protective effect of family support and empathy, school achievement, and school attachment was observed for both bullying and cyberbullying perpetration in this study. Results indicated that there was a substantial decrease in the proportion of bullies and cyberbullies as parent support and level of empathy increased, and these factors contributed the most to reducing the risk of both behaviors. This is consistent with research suggesting that a positive home environment, good parental communication, involvement, and family support are important in bullying and cyberbullying prevention [31,36,40,41,42]. Support from parents offers the strongest protection against bullying and cyberbullying perpetration, and good communication and relationships are the most significant. Parental support and attachment have been determined to be the strongest factors that can protect adolescents from perpetration of bullying and cyberbullying [62]. Subsequently, a strong protective effect of empathy in terms of protection from becoming a bully and a cyberbully was confirmed as well, and the effect sizes of these relationships proved to be significant. Reviews and a meta-analysis of the studies on the relationships between empathy and bullying and cyberbullying proved that empathy is strongly related to both aggressive behaviors [4,29]. This study revealed a significant relationship between empathy and both forms of perpetration, and this might have an important implication for prevention programs. 

Our study has also confirmed the preventive effect of school performance. This is in line with the results from other studies that show that a high level of academic achievement is a strong protective factor against bullying and cyberbullying perpetration. Good academic performance has been found to be related to low bullying and cyberbullying perpetration [29,31]. Additionally, school attachment as a sense of belonging to the school is a significant protective factor against bullying and cyberbullying perpetration. It has been shown that positive school commitment and school climate were related to low rates of being a bully [63]. Studies showed that bullies and cyberbullies did not like school, did not feel safe and cared for by teachers, and a felt weak sense of belonging to their school [31,64,65].

Life satisfaction as a protective factor was identified only in relation to cyberbullying and not in terms of perpetration of in-person bullying. Our findings are partially in line with other studies, which show that life satisfaction and psychological wellbeing were related to both low in-person and cyberbullying perpetration [33,66]. 

Social self-efficacy turned out to be an insignificant predictor, which was in opposition to other studies, which showed that both perpetration and cyber-perpetration were negatively associated with social self-efficacy [33,67]. However, results regarding self-efficacy vary, and other studies indicated that firmer self-efficacy beliefs are positively associated with high rates of cyberbullying [68].

Our study did not confirm relationships between perceived peer support and either form of bullying. Regarding peer protective factors against bullying and cyberbullying, besides peer support, studies emphasize the role of high peer status, popularity, positive influence, and perceived peer normative beliefs against aggression [29,31]. Therefore, more of the identified peer factors related to aggression should be considered to find the protective impacts.

Beyond looking at the relative magnitude of effect sizes within the groups of bullies and cyberbullies, we were interested in assessing them comparatively across groups. Thus, we tried to answer the question of whether bullies and cyberbullies differed in psychosocial determinants and, if so, to what extent. The straightforward answer to this question is that both shared and unique predictors were observed for bullies and cyberbullies. 

Boys appeared to be more involved in both bullying and cyberbullying than girls. The impact of age and family structure was assessed as well for both forms of bullying. Younger pupils and ones from intact families perpetrated both bullying and cyberbullying less often. Family support and school attachment protected against involvement in perpetration to a significant extent, indicating the important role of the home and school environment in the prevention of bullying and cyberbullying. Empathy and school performance were an additional common protective predictor across these two groups, and these individual factors protect adolescents from involvement in both bullying and cyberbullying. Among individual factors, life satisfaction was found to be a unique predictor that distinguished cyberbullies from in-person bullies. Finally, as far as socioeconomic factors are concerned, a richer region of the place of residence protects from perpetration of in-person bullying, whereas high family affluence was a risk factor for cyberbullying perpetration only.

The present study was limited to measuring both bullying and cyberbullying perpetration using a single item for each. These two items available in the HBSC protocol concerning bullying and cyberbullying perpetration were used, a dichotomous division was conducted, and two clusters were then defined based on the responses. This measure has not been developed enough to assess the specific forms, severity, frequency, and situational circumstances related to bullying or cyberbullying. An analysis of the data regarding more specific forms and circumstances related to perpetration would be worthwhile to carry out.

The cross-sectional nature of this study limits the interpretation of the directionality of the associations between psychosocial factors and bullying. It makes causal inferences on relevant predictors and bullying impossible; therefore, our findings do not provide grounds for conclusions based on cause and effect [69]. Studies with a longitudinal design are needed to address this gap and to untangle the relationships between bullying and its psychosocial determinants. Moreover, longitudinal studies are recommended for examining the differences between bullies and cyberbullies to provide clarity on the significance and impact of predictors that protect youth from aggressive behaviors throughout their development.

There are some limitations regarding other important factors included in the SEM since this was not available in the collected data. Future studies need to include a broader set of SEM factors that may decrease the risk of in-person and cyberbullying, such as family environment (parental control and discipline), social school climate (school safety), and broader community and regional contexts. Furthermore, regarding cyberbullying, the specific situational factors relevant to the online context should be included (online social network and social media use, technical skills, individual awareness of cyber violence risks) and need to be considered in future studies.

Nevertheless, it seems that a number of other factors in this study offset the above limitations. One of them is the large representative study sample, consisting of almost 3650 adolescents aged 15–17. The subject of the analysis was the incidence and selected determinants of bullying and cyberbullying perpetration. A standardized questionnaire was used, and the scales were previously verified in other studies. To date, several national and international papers on peer violence have been published based on the results of HBSC research from 2018 [18]. They mainly concern teenagers aged 11–15. The advantage of the research conducted in Poland is the addition of another, older group, which provides a picture of changes occurring after the age of 15 years. Some data related to potential psychosocial determinants could be taken into analysis from the questions which were asked only to older youth (aged 15 and 17). As Poland is a country with high rates of bullying and cyberbullying as compared with the HBSC ranking, assessment of the protective psychosocial factors seems to be useful for peer violence prevention targeting older adolescents.

## 5. Conclusions

Based on the current results, tailored anti-bullying and anti-cyberbullying programs with components focused not only on the individual and social protective factors but also targeting socially and economically deprived regions should be implemented in Polish schools.

This study demonstrated that certain socioeconomic, individual, family, and school-related factors similarly or differentially contribute to protection against becoming a bully or cyberbully, suggesting the importance of including both in prevention programs. These include development of emotional and interpersonal competencies of the students, supporting their school achievements, improving communication and relationships in the family, and developing a positive school social climate. Approaching life satisfaction as a protective factor against cyberbullying may require unique strategies that consider the well-being of adolescents. The study accentuates the important protective role of family and school. The impact of positive feelings toward school suggests the importance of a school climate where all students feel connected, cared for, and respected. 

Bullying and cyberbullying perpetration is a complex problem to solve, which requires a comprehensive and multidimensional approach in youth violence prevention policies. Peer violence prevention school programs targeting at risk adolescents are not sufficient in Poland. Prevention of discriminatory (towards LGBT+ groups, ethnic and national minorities, migrants, as well as those who experience poverty and socio-economic exclusion) needs a stronger strategic focus. The social-ecological perspective may be effective in addressing these issues. Reviews of bullying and cyberbullying prevention programs highlight some key features of successful interventions [5,38]. Findings of our study support the approach to include anti-bullying and anti-cyberbullying school programs within a safe school context, with a crucial role of school leaders and teachers to strengthen adolescent health and well-being [70,71]. A common strategy in bullying and cyberbullying prevention may be useful, including improving the school social climate, family and individual student outreach support, and the development of emotional, cognitive, and social skills in education curriculums. Moreover, specific national strategies targeting socially and economically deprived regions should be considered in adolescent violence prevention programs.

## Figures and Tables

**Table 1 ijerph-20-01358-t001:** Bullying and cyberbullying perpetration at least once in the past two months (%) according to demographic and socioeconomic factors.

Variable	Sample *n* (%)	Bullying	Chi-sq *p*	Cyberbullying	Chi-sq *p*
Gender					
Boys	1715 (47.0)	27.7	60.87	22.9	71.78
Girls	1935 (53.0)	16.9	<0.001	12.3	<0.001
Grade					
K9	1955 (53.6)	26.2	43.82	19.3	12.80
K11	1695 (46.4)	17.1	<0.001	14.8	<0.001
Family structure					
Non-intact	900 (24.7)	25.4	8.26	20.4	8.24
Intact	2750 (75.3)	20.8	0.004	16.2	0.004
FAS					
Low	1022 (28.4)	23.0	1.25	16.1	1.54
Average	1718 (47.8)	21.8	0.535	17.4	0.463
High	855 (23.8)	20.9		18.1	
LD quintiles					
Q1—poorest	486 (13.3)	25.3		19.8	
Q2	488 (13.4)	25.5	17.34	17.2	2.71
Q3	603 (16.5)	22.9	0.002	16.7	0.608
Q4	670 (18.4)	23.4		16.7	
Q5—richest	1403 (38.4)	18.5		16.8	

FAS—family affluence scale; LD—local deprivation.

**Table 2 ijerph-20-01358-t002:** Participation in bullying and cyberbullying as perpetrators at least once in the past two months (%) according to individual factors.

Variable	Sample *n* (%)	Bullying	Chi-sq *p*	Cyberbullying	Chi-sq *p*
School performance					
Low	706 (19.5)	27.2	14.60	21.7	15.79
Average	1957 (54.0)	21.3	0.001	17.1	<0.001
High	962 (26.5)	19.7		14.3	
Social self-efficacy					
Low	855 (24.0)	24.1	3.24	19.7	6.76
Average	1914 (53.7)	21.2	0.198	17.2	0.034
High	793 (22.3)	21.1		14.8	
Empathy					
Low	631 (17.5)	33.1	67.74	26.2	53.33
Average	2126 (59.0)	20.9	<0.001	16.8	<0.001
High	849 (23.5)	15.5		11.8	
Life satisfaction					
Low	767 (21.2)	24.7	4.31	22.3	18.25
Average	2114 (58.3)	21.0	0.116	16.0	<0.001
High	745 (20.5)	21.9		15.2	

**Table 3 ijerph-20-01358-t003:** Participation in bullying and cyberbullying as perpetrators at least once in the past two months (%) according to social factors.

Variable	Sample *n* (%)	Bullying	Chi-sq *p*	Cyberbullying	Chi-sq *p*
Family support					
Low	872 (25.3)	26.2	22.01	21.7	27.79
Average	1771 (51.4)	22.2	<0.001	17.5	<0.001
High	804 (23.3)	16.8		12.0	
Peer support					
Low	785 (21.7)	29.3	27.79	21.0	12.15
Average	1970 (54.3)	20.3	<0.001	17.0	0.002
High	869 (24.0	19.0		14.6	
School attachment					
Very low	422 (11.6)	30.6		28.7	
Rather low	732 (20.1)	24.7	29.39	16.1	44.40
Average or high	2481 (68.3)	19.7	<0.001	15.6	<0.001

**Table 4 ijerph-20-01358-t004:** The correlations of psychosocial determinants of bullying and cyberbullying perpetration. (Spearman’s rho).

Variables	1	2	3	4	5	6	7	8	9
1. FAS	1	0.192 **	0.109 **	0.123 **	0.033 *	0.153 **	0.100 **	0.060 **	0.018
2. LD		1	0.023	0.042 *	0.083 **	−0.026	−0.028	0.037 *	0.028
3. SP			1	0.110 **	0.085 **	0.219 **	0.173 **	0.117 **	0.153 **
4. SSE				1	0.181 **	0.238 **	0.281 **	0.442 **	0.193 **
5. EMP					1	−0.032	0.057 **	0.191 **	0.084 **
6. LS						1	0.415**	0.231 **	0.254 **
7. FSUP							1	0.310 **	0.206 **
8. PSUP								1	0.286 **
9. SA									1

FAS—family affluence scale; LD—local deprivation; SP—school performance; SSE—social self-efficacy; EMP—empathy; LS—life satisfaction; FSUP—family support; PSUP—peer support; SA—school attachment; rho above diagonal; * *p* < 0.05; ** *p* < 0.001.

**Table 5 ijerph-20-01358-t005:** Logistic regression predicting likelihood of bullying perpetration.

Independent Variables	B	S.E.	*p*	Odds Ratio	95% CI for Odds Ratio	
					Lower	Upper
Gender (ref. girls)	0.661	0.094	<0.001	1.937	1.612	2.327
Grade (ref. older)	0.570	0.093	<0.001	1.769	1.474	2.123
Family structure (ref. intact)	0.207	0.101	0.041	1.230	1.009	1.500
Family affluence (ref. low)			0.944			
Family affluence average	−0.030	0.104	0.777	0.971	0.791	1.191
Family affluence high	−0.039	0.128	0.758	0.961	0.748	1.236
Deprivation index (ref. richest Q5)			0.017			
Deprivation index poorest (Q1)	0.315	0.140	0.024	1.370	1.041	1.803
Deprivation index (Q2)	0.416	0.142	0.004	1.515	1.146	2.003
Deprivation index (Q3)	0.256	0.132	0.052	1.292	0.997	1.674
Deprivation index (Q4)	0.297	0.127	0.019	1.346	1.050	1.725
School performance (ref. high)			0.116			
School performance low	0.278	0.135	0.040	1.321	1.013	1.723
School performance average	0.105	0.109	0.336	1.111	0.897	1.376
Social self-efficacy (ref. high)			0.156			
Social self-efficacy low	−0.270	0.143	0.059	0.764	0.577	1.010
Social self-efficacy average	−0.174	0.117	0.138	0.840	0.668	1.057
Empathy (ref. high)			<0.001			
Empathy low	0.627	0.145	<0.001	1.873	1.408	2.490
Empathy average	0.302	0.120	0.012	1.352	1.068	1.712
Life satisfaction (ref. high)			0.564			
Life satisfaction low	0.005	0.152	0.974	1.005	0.746	1.353
Life satisfaction average	−0.096	0.120	0.421	0.908	0.718	1.148
Family support (ref. high)			<0.001			
Family support low	0.628	0.146	<0.001	1.873	1.407	2.493
Family support average	0.416	0.125	<0.001	1.516	1.186	1.936
Peer support (ref. high)			0.020			
Peer support low	0.114	0.145	0.430	1.121	0.844	1.488
Peer support average	−0.182	0.119	0.125	0.833	0.660	1.052
School attachment (ref. average or high)			0.011			
School attachment very low	0.388	0.138	0.005	1.474	1.125	1.932
School attachment rather low	0.203	0.112	0.071	1.225	0.983	1.528
Constant	−2.790	0.222	<0.001	0.061		
R-Sq Nagelkerke	0.096					

**Table 6 ijerph-20-01358-t006:** Logistic regression predicting likelihood of cyberbullying perpetration.

Independent Variables	B	S.E.	*p*	Odds Ratio	95% CI for Odds Ratio	
					Lower	Upper
Gender (ref. girls)	0.779	0.103	<0.001	2.180	1.780	2.670
Grade (ref. older)	0.345	0.101	<0.001	1.413	1.159	1.721
Family structure (ref. intact)	0.269	0.109	0.013	1.309	1.057	1.621
Family affluence (ref. low)			0.066			
Family affluence average	0.196	0.117	0.094	1.217	0.967	1.531
Family affluence high	0.320	0.140	0.022	1.377	1.047	1.812
Deprivation index (ref. richest Q5)			0.912			
Deprivation index poorest (Q1)	0.112	0.154	0.467	1.119	0.827	1.512
Deprivation index (Q2)	0.067	0.158	0.673	1.069	0.784	1.457
Deprivation index (Q3)	0.119	0.143	0.406	1.126	0.851	1.491
Deprivation index (Q4)	0.081	0.138	0.559	1.084	0.827	1.422
School performance (ref. high)			0.128			
School performance low	0.299	0.149	0.044	1.349	1.008	1.805
School performance average	0.171	0.122	0.159	1.187	0.935	1.506
Social self-efficacy (ref. high)			0.674			
Social self-efficacy low	−0.024	0.158	0.879	0.976	0.716	1.330
Social self-efficacy average	0.074	0.132	0.574	1.077	0.832	1.394
Empathy (ref. high)			<0.001			
Empathy low	0.622	0.161	<0.001	1.863	1.359	2.554
Empathy average	0.363	0.135	0.007	1.437	1.104	1.871
Life satisfaction (ref. high)			0.010			
Life satisfaction low	0.352	0.164	0.032	1.422	1.031	1.962
Life satisfaction average	−0.029	0.134	0.826	0.971	0.747	1.262
Family support (ref. high)			<0.001			
Family support low	0.647	0.160	<0.001	1.909	1.396	2.612
Family support average	0.381	0.139	0.006	1.464	1.114	1.923
Peer support (ref. high)			0.629			
Peer support low	−0.125	0.161	0.436	0.882	0.644	1.209
Peer support average	−0.122	0.130	0.349	0.885	0.685	1.143
School attachment (ref. average or high)			<0.001			
School attachment very low	0.636	0.142	<0.001	1.888	1.430	2.493
School attachment rather low	−0.121	0.129	0.350	0.886	0.688	1.142
Constant	−3.427	0.251	<0.001	0.032		
R-Sq Nagelkerke	0.094					

## Data Availability

The data presented in this study are available on request from the corresponding author. The data are not publicly available due to the internal HBSC data access policy. Data access to the HBSC rounds is provided by the HBSC Data Management Centre—Department of Health Promotion and Development, University of Bergen (https://www.uib.no/en/hbscdata, accessed on 21 November 2022).

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
