# Peer review of "Similarities and Differences between Psychosocial Determinants of Bullying and Cyberbullying Perpetration among Polish Adolescents"

_ijerph, 2023, doi:10.3390/ijerph20021358_

Round 1
Reviewer 1 Report
1. Fine-tune the title and the abstract to highlight the key objective of this research.
2. Clarify the research is focusing on the status of Poland while introducing this study in beginning.
3. The authors prefer to showcase the differences between bullying and cyber-bullying, or represent the factors influencing them differently? As described in the title, it sounds that you may only want to find out if bullying and cyber-bullying is different?
Overall, the paper is not about the question of Yes or No. It should make research on How and Why. You actually did so, but the title, the abstract and the introduction could not precisely reflect the research questions and its objective.
4. It seems that SEM is very important for this study, but it's introduced so late in the introduction, with too many words description. Why not use some simple table or figure to show it clearly?
5. Similarly, too many words on so many descriptions. The questionnaire, the factors, the predictors, the result... It makes reading and understanding more or less hard therefore wasting the readers' time. The paper is valuable and should not be very complex, whereas this study is mainly based on a questionnaire. But it's represented so complicated with many long-winded words.
Please leverage more qualified tables or figures to summarize or illustrate those words description except for the tables regarding questionnaire.
6. Finally, I have to say, the questionnaire methodology in this study may be effective on describing or explaining the status of adolescent perpetrators of bullying and cyber-bullying, but less effective on explaining or exploring the differences between them.
Wish the authors continue in-depth research and share more valuable findings in future.
Author Response
Please see the attachment.
Response to Reviewer 1 Comments
Thank you for your kind and constructive comments and feedback. Kindly find our response to the comments below.
Point 1: Fine-tune the title and the abstract to highlight the key objective of this research.
Response 1: We changed the title and improved the abstract to highlight the key objective of this research.
Point 2: Clarify the research is focusing on the status of Poland while introducing this study in beginning.
Response 2: We clarified it in the title.
Point 3: The authors prefer to showcase the differences between bullying and cyber-bullying, or represent the factors influencing them differently? As described in the title, it sounds that you may only want to find out if bullying and cyber-bullying is different? Overall, the paper is not about the question of Yes or No. It should make research on How and Why. You actually did so, but the title, the abstract and the introduction could not precisely reflect the research questions and its objective.
Response 3: We have rewritten the title and the goal in the abstract, and we hope it explains more clear that we aimed to show similarities and differences in selected predictors of peer violence perpetration among adolescents. The aim was to asses if the factors influencing these two aggressive behaviors differently, so we changed and underlined that this study is about the question of how psychosocial determinates of bullying and cyberbullying differ.
Point 4: It seems that SEM is very important for this study, but it's introduced so late in the introduction, with too many words description. Why not use some simple table or figure to show it clearly?
Response 4: We have referred to the SEM as the public health perspective in understanding determinants of violence, including adolescent peer violence. It assumes specific socioeconomic, individual, and social determinants, and we have chosen it as the theoretical foundation in the analysis. However we did not intent to analyze and discuss the model, so we limited this part.
Point 5: Similarly, too many words on so many descriptions. The questionnaire, the factors, the predictors, the result... It makes reading and understanding more or less hard therefore wasting the readers' time. The paper is valuable and should not be very complex, whereas this study is mainly based on a questionnaire. But it's represented so complicated with many long-winded words. Please leverage more qualified tables or figures to summarize or illustrate those words description except for the tables regarding questionnaire.
Response 5: We have rewritten the results part to summarize and synthesis the findings showed in the tables. We reduced and simplified the comments to make it more clear in the text.
Point 6: Finally, I have to say, the questionnaire methodology in this study may be effective on describing or explaining the status of adolescent perpetrators of bullying and cyber-bullying, but less effective on explaining or exploring the differences between them.
Response 6: We included in the study limitations part, that the survey method applied in our study seems to be effective on describing or explaining the status of bullies and cyberbullies in the context of demographic, socio-economic, individual and social factors, but very limited in explaining causal relationships or exploring the differences between them. We hope to continue the research in bullying and cyberbullying more in-depth studies, and share more valuable findings in future possible to apply in school violence prevention programs.
We appreciate your suggestions that our manuscript should be improved in English editing and style. We corrected the text. The article has been checked by professional English proofreading services company.

Reviewer 2 Report
In the Introduction should be more information about some perpetrators involved in bullying and cyberbullying, such as witnesses and passive aggressors.
In the Conclusions, in addition to the need for bullying and cyberbullying prevention programs, I miss some specific actions, for exemple, by involving school leaders, actions could be taken to reduce violent situations in schools.
Author Response
Please see the attachment.
Response to Reviewer 2 Comments
Thank you for your kind and constructive comments and feedback. Kindly find our response to the comments below.
Point 1: In the Introduction should be more information about some perpetrators involved in bullying and cyberbullying, such as witnesses and passive aggressors.
Response 1: We agree that in analysis of complex adolescent peer violence, the data analysis regarding witnesses (bystanders) and passive aggressors (bully-victims) are very important to complete the full picture of the problem. However, we narrowed our analysis to two groups, i.e. in-person bullies and cyberbullies, to assess to what extent the impacts of selected psychosocial determinants of adolescents peer violence differ in these two groups. In next studies, we would like to consider the passive aggressors (bully-victims). Unfortunately, data on witnesses (bystanders) were not collected in this survey.
Point 2: In the Conclusions, in addition to the need for bullying and cyberbullying prevention programs, I miss some specific actions, for exemple, by involving school leaders, actions could be taken to reduce violent situations in schools.
Response 2: Findings of our study support the approach to include anti-bullying and anti-cyberbullying school programs within a safe school context, with crucial role of school leaders and teachers to strength adolescent health and wellbeing. In Conclusions we added the sentence and two more references [70,71].
